

# The earth is flat ($p > 0.05$): significance thresholds and the crisis of unreplicable research

Valentin Amrhein[1,2,3], Fränzi Korner-Nievergelt[3,4] and Tobias Roth[1,2]

[1] Zoological Institute, University of Basel, Basel, Switzerland
[2] Research Station Petite Camargue Alsacienne, Saint-Louis, France
[3] Swiss Ornithological Institute, Sempach, Switzerland
[4] Oikostat GmbH, Ettiswil, Switzerland

Corresponding author
Valentin Amrhein,
v.amrhein@unibas.ch

## ABSTRACT

The widespread use of 'statistical significance' as a license for making a claim of a scientific finding leads to considerable distortion of the scientific process (according to the American Statistical Association). We review why degrading $p$-values into 'significant' and 'nonsignificant' contributes to making studies irreproducible, or to making them seem irreproducible. A major problem is that we tend to take small $p$-values at face value, but mistrust results with larger $p$-values. In either case, $p$-values tell little about reliability of research, because they are hardly replicable even if an alternative hypothesis is true. Also significance ($p \leq 0.05$) is hardly replicable: at a good statistical power of 80%, two studies will be 'conflicting', meaning that one is significant and the other is not, in one third of the cases if there is a true effect. A replication can therefore not be interpreted as having failed only because it is nonsignificant. Many apparent replication failures may thus reflect faulty judgment based on significance thresholds rather than a crisis of unreplicable research. Reliable conclusions on replicability and practical importance of a finding can only be drawn using cumulative evidence from multiple independent studies. However, applying significance thresholds makes cumulative knowledge unreliable. One reason is that with anything but ideal statistical power, significant effect sizes will be biased upwards. Interpreting inflated significant results while ignoring nonsignificant results will thus lead to wrong conclusions. But current incentives to hunt for significance lead to selective reporting and to publication bias against nonsignificant findings. Data dredging, $p$-hacking, and publication bias should be addressed by removing fixed significance thresholds. Consistent with the recommendations of the late Ronald Fisher, $p$-values should be interpreted as graded measures of the strength of evidence against the null hypothesis. Also larger $p$-values offer some evidence against the null hypothesis, and they cannot be interpreted as supporting the null hypothesis, falsely concluding that 'there is no effect'. Information on possible true effect sizes that are compatible with the data must be obtained from the point estimate, e.g., from a sample average, and from the interval estimate, such as a confidence interval. We review how confusion about interpretation of larger $p$-values can be traced back to historical disputes among the founders of modern statistics. We further discuss potential arguments against removing significance thresholds, for example that decision rules should rather be more stringent, that sample sizes could decrease, or that $p$-values should better be completely abandoned. We conclude that

whatever method of statistical inference we use, dichotomous threshold thinking must give way to non-automated informed judgment.

# INTRODUCTION

"It seems to me that statistics is often sold as a sort of alchemy that transmutes randomness into certainty, an 'uncertainty laundering' that begins with data and concludes with success as measured by statistical significance. (...) The solution is not to reform *p*-values or to replace them with some other statistical summary or threshold, but rather to move toward a greater acceptance of uncertainty and embracing of variation."

*Gelman (2016)*

Scientific results can be irreproducible for at least six major reasons (*Academy of Medical Sciences, 2015*). There may be (1) technical problems that are specific to the particular study. There may be more general problems like (2) weak experimental design or (3) methods that are not precisely described so that results cannot be reproduced. And there may be statistical issues affecting replicability that are largely the same in many fields of research. Such issues are (4) low statistical power, and (5) 'data dredging' or '*p*-hacking' by trying alternative analyses until a significant result is found, which then is selectively reported without mentioning the nonsignificant outcomes. Related to that, (6) publication bias occurs when papers are more likely to be published if they report significant results (*Bishop & Thompson, 2016*).

Is a major part of an apparent crisis of unreplicable research caused by the way we use statistics for analyzing, interpreting, and communicating our data? Significance testing has been severely criticized for about a century (e.g., *Boring, 1919*; *Berkson, 1938*; *Rozeboom, 1960*; *Oakes, 1986*; *Cohen, 1994*; *Ziliak & McCloskey, 2008*; *Kline, 2013*), but the prevalence of *p*-values in the biomedical literature is still increasing (*Chavalarias et al., 2016*). For this review, we assume that a revolution in applied statistics with the aim of banning *p*-values is not to be expected nor necessarily useful, and that the main problem is not *p*-values but how they are used (*Gelman, 2013b*; *Gelman, 2016*). We argue that one of the smallest incremental steps to address statistical issues of replicability, and at the same time a most urgent step, is to remove thresholds of statistical significance like $p = 0.05$ (see Box 1). This may still sound fairly radical to some, but for the following reasons it is actually not.

First, *p*-values can be traditionally employed and interpreted as evidence against null hypotheses also without using a significance threshold. However, what needs to change for reducing data dredging and publication bias is our overconfidence in what significant *p*-values can tell, and, as the other side of the coin, our bad attitude towards *p*-values that do not pass a threshold of significance. As long as we treat our larger *p*-values as unwanted children, they will continue disappearing in our file drawers, causing publication bias, which

has been identified as the possibly most prevalent threat to reliability and replicability of research already a long time ago (*Sterling, 1959*; *Wolf, 1961*; *Rosenthal, 1979*). Still today, in an online survey of 1576 researchers, selective reporting was considered the most important factor contributing to irreproducible research (*Baker, 2016*).

Second, the claim to remove fixed significance thresholds is widely shared among statisticians. In 2016, the American Statistical Association (ASA) published a statement on *p*-values, produced by a group of more than two dozen experts (*Wasserstein & Lazar, 2016*). While there were controversial discussions about many topics, the consensus report of the ASA features the following statement: "The widespread use of 'statistical significance' (generally interpreted as '$p \leq 0.05$') as a license for making a claim of a scientific finding (or implied truth) leads to considerable distortion of the scientific process" (*Wasserstein & Lazar, 2016*). And a subgroup of seven ASA statisticians published an extensive review of 25 misinterpretations of *p*-values, confidence intervals, and power, closing with the words: "We join others in singling out the degradation of *p*-values into 'significant' and 'nonsignificant' as an especially pernicious statistical practice" (*Greenland et al., 2016*).

The idea of using *p*-values not as part of a binary decision rule but as a continuous measure of evidence against the null hypothesis has had many advocates, among them the late Ronald Fisher (*Fisher, 1956*; *Fisher, 1958*; *Eysenck, 1960*; *Skipper, Guenther & Nass, 1967*; *Labovitz, 1968*; *Edgington, 1970*; *Oakes, 1986*; *Rosnow & Rosenthal, 1989*; *Stoehr, 1999*; *Sterne & Smith, 2001*; *Gelman, 2013a*; *Greenland & Poole, 2013*; *Higgs, 2013*; *Savitz, 2013*; *Madden, Shah & Esker, 2015*; *Drummond, 2016*; *Lemoine et al., 2016*; *Van Helden, 2016*). Removing significance thresholds was also suggested by authors sincerely defending *p*-values against their critics (*Weinberg, 2001*; *Hurlbert & Lombardi, 2009*; *Murtaugh, 2014a*).

In the following, we start with reviewing what *p*-values can tell about replicability and reliability of results. That this will not be very encouraging should not be taken as another advice to stop using *p*-values. Rather, we want to stress that reliable information about reliability of results cannot be obtained from *p*-values nor from any other statistic calculated in individual studies. Instead, we should design, execute, and interpret our research as a 'prospective meta-analysis' (*Ioannidis, 2010*), to allow combining knowledge from multiple independent studies, each producing results that are as unbiased as possible. Our aim is to show that not *p*-values, but significance thresholds are a serious obstacle in this regard.

We therefore do not focus on general misconceptions about *p*-values, but on problems with, history of, and solutions for applying significance thresholds. After discussing why significance cannot be used to reliably judge the credibility of results, we review why applying significance thresholds reduces replicability. We then describe how the switch in interpretation that often follows once a significance threshold is crossed leads to proofs of the null hypothesis like 'the earth is flat ($p > 0.05$)'. We continue by summarizing opposing recommendations by Ronald Fisher versus Jerzy Neyman and Egon Pearson that led to the unclear status of nonsignificant results, contributing to publication bias. Finally, we outline how to use graded evidence and discuss potential arguments against removing significance thresholds. We conclude that we side with a neoFisherian paradigm of treating

*p*-values as graded evidence against the null hypothesis. We think that little would need to change, but much could be gained by respectfully discharging significance, and by cautiously interpreting *p*-values as continuous measures of evidence.

---

**Box 1.**    Significance thresholds and two sorts of reproducibility

**Inferential reproducibility** might be the most important dimension of reproducibility and "refers to the drawing of qualitatively similar conclusions" from an independent replication of a study (*Goodman, Fanelli & Ioannidis, 2016*). Some people erroneously conclude that a nonsignificant replication automatically contradicts a significant original study. Others will look at the observed effect, which might hint into the same direction as in the original study, and therefore see no contradiction. Since judgment based on significance is faulty, judgment based on effect sizes will increase inferential reproducibility. Further, it is current practice to interpret *p*-values >0.05 either as a statistical trend, or (falsely) as evidence in favor of a null effect, or as no evidence at all. Researchers will increase inferential reproducibility if they refrain from turning their conclusion upside down once a significance threshold is crossed, but instead take the *p*-value as providing graded evidence against the null hypothesis.

**Results reproducibility**, or **replicability**, "refers to obtaining the same results from the conduct of an independent study" (*Goodman, Fanelli & Ioannidis, 2016*). How results should look like to be considered 'the same', however, remains operationally elusive. What matters, according to *Goodman, Fanelli & Ioannidis (2016)*, "is not replication defined by the presence or absence of statistical significance, but the evaluation of the cumulative evidence and assessment of whether it is susceptible to major biases". Unfortunately, adhering to significance thresholds brings considerable bias to the published record of cumulative evidence. If results are selected for publication or interpretation because they are significant, conclusions will be invalid. One reason is that the lens of statistical significance usually sees only inflated effects and results that are "too good to be true" (*Gelman, 2015*). Researchers will increase replicability if they report and discuss all results, irrespective of the sizes of their *p*-values.

---

### *P*-values are hardly replicable

In most cases, null hypothesis significance testing is used to examine how compatible some data are with the null hypothesis that the true effect size is zero. The statistical test result is a *p*-value informing on the probability of the observed data, or data more extreme, given that the null hypothesis is true (and given that all other assumptions about the model are correct; *Greenland et al., 2016*). If $p \leq 0.05$, we have learned in our statistics courses to call this significant, to reject the null hypothesis, and to accept an alternative hypothesis about some non-zero effect in the larger population.

However, we do not know nor can we infer whether the null hypothesis or an alternative hypothesis is true. On the basis of one single study, it is logically impossible to draw a firm conclusion (*Oakes, 1986*, p. 128); for example, because a small *p*-value either means the null hypothesis is not true, or else it is true but we happened to find relatively unlikely data. It is

for those "possible effects of chance coincidence" that Ronald Fisher wrote: "No isolated experiment, however significant in itself, can suffice for the experimental demonstration of any natural phenomenon" (*Fisher, 1937*, p. 16).

Unlike a widespread belief, the *p*-value itself does not indicate how replicable our results are (*Miller, 2009*; *Greenland et al., 2016*). We hope that a small *p*-value means our results are reliable and a replication study would have a good chance to find a small *p*-value again. Indeed, an extensive research project replicating 100 psychological studies reported that the chance to find a significant result in a replication was higher if the *p*-value in the original study was smaller; but of 63 original studies with $p < 0.02$, only 26 (41%) had $p < 0.05$ in the replication (*Open Science Collaboration, 2015*).

Apparently, *p*-values are hardly replicable. This is most evident if the null hypothesis is true, because then *p*-values are uniformly distributed and thus all values are equally likely to occur (*Hung et al., 1997*; *Boos & Stefanski, 2011*; *Colquhoun, 2014*). However, a null hypothesis of an effect of exactly zero is often unlikely to be true (*Loftus, 1993*; *Cohen, 1994*; *Stahel, 2016*). After all, in most cases we did our study because we had some *a priori* reason to believe that the true effect is not zero.

Unfortunately, *p*-values are highly variable and thus are hardly replicable also if an alternative hypothesis is true, or if the observed effect size is used for calculating the distribution of *p* (*Goodman, 1992*; *Senn, 2002*; *Cumming, 2008*; *Halsey et al., 2015*). *Cumming (2008)* showed that if we observe an effect with $p = 0.05$ in a first study, a replication study will find a *p*-value between 0.00008 and 0.44 with 80% probability (given by an 80% 'prediction interval'), and of >0.44 with 10% probability. For $p = 0.5$ in a first study, *Lazzeroni, Lu & Belitskaya-Levy (2014)* found that 95% of replication *p*-values will have sizes between 0.003 and 0.997 (given by a 95% prediction interval).

This enormous variability from sample to sample was called the 'dance of the *p*-values' (*Cumming, 2012*; *Cumming, 2014*). Because the *p*-value is based upon analysis of random variables, it is a random variable itself, and it behaves as such (*Hung et al., 1997*; *Sackrowitz & Samuel-Cahn, 1999*; *Murdoch, Tsai & Adcock, 2008*). For some reason, however, the stochastic aspect of *p*-values is usually neglected, and *p* is reported as a fixed value without a measure of vagueness or unreliability (*Sackrowitz & Samuel-Cahn, 1999*; *Cumming, 2008*; *Barber & Ogle, 2014*). Indeed, we cannot use standard errors or confidence intervals for *p*, because they would estimate unobservable population parameters; and because the *p*-value is a property of the sample, there is no unobservable 'true *p*-value' in the larger population (*Cumming, 2012*, p. 133). But as shown, e.g., by *Cumming (2008)*, we could present our *p*-values with prediction intervals, which are intervals with a specified chance of including the *p*-value given by a replication.

If we would make vagueness of *p*-values visible by using prediction intervals, it would become immediately apparent that the information content of $p = 0.04$ and of $p = 0.06$ is essentially the same (*Dixon, 2003*; *Halsey et al., 2015*; *Giner-Sorolla, 2016*), and that "the difference between 'significant' and 'not significant' is not itself statistically significant"

(*Gelman & Stern, 2006*). It is a good habit to publish exact *p*-values rather than uninformative statements like '$p > 0.05$'; but additional decimal places and an equal sign should not mislead us to give *p*-values an aura of exactitude (*Boos & Stefanski, 2011*; *Halsey et al., 2015*).

*P*-values are only as reliable as the samples from which they are obtained (*Halsey et al., 2015*). They inherit their vagueness from the uncertainty of point estimates like the sample average from which they are calculated. But they clearly give less information on uncertainty, reliability or replicability of the point estimate than is evident from a 95% confidence interval (which is an 83% prediction interval for the point estimate of a replication; *Cumming, 2014*). While the confidence interval measures precision and, therefore, reliability of the point estimate, the *p*-value mixes information on the size of the effect and how precisely it was measured. Thus, two point estimates can be equally reliable but may have different effect sizes and therefore different *p*-values (Figs. 1A, 1D). And a small *p*-value can arise because a point estimate is far off the null value, but data may still show considerable variation around the point estimate that therefore would not be very reliable (Figs. 1A versus 1E).

So by definition, the *p*-value reflects our observed evidence against a null hypothesis, but it does not directly measure reliability of the effect that we found in our sample. And we saw that *p*-values are much more variable than most people think (*Lai, Fidler & Cumming, 2012*). We therefore must learn to treat *p*-values like any other descriptive statistic and refrain from taking them at face value when we want to draw inference beyond our particular sample data (*Miller, 2009*). Using observed *p*-values to make a binary decision whether or not to reject a hypothesis is as risky as placing our bets on a sample average without considering that there might be error attached to it. If *p*-values are hardly replicable, so too are decisions based on them (*Kline, 2013*, p. 13).

It seems that the only way to know how replicable our results are is to actually replicate our results. Science will proceed by combining cumulative knowledge from several studies on a particular topic, summarized for example in meta-analyses (*Schmidt, 1992*; *Schmidt, 1996*; *Goodman, Fanelli & Ioannidis, 2016*). And one reason why replication studies are rarely done (*Kelly, 2006*) may be that between 37% and 60% of academic professionals seem to think the *p*-value informs on the probability of replication (*Gigerenzer, Krauss & Vitouch, 2004*). After all, why actually replicate a study when the *p*-value gives us virtual replications (*Ziliak & McCloskey, 2008*, p. 127)?

If we do a true replication study, however, our observed *p*-value will be a realization of a random variable again and will be as unreliable as in the first study. A single replication thus can neither validate nor invalidate the original study (*Maxwell, Lau & Howard, 2015*; *Leek & Jager, 2017*; *Nosek & Errington, 2017*). It simply adds a second data point to the larger picture.

## Significance is hardly replicable

There is currently no consensus how replicability or results reproducibility should be measured (see Box 1; *Open Science Collaboration, 2015*; *Baker, 2016*; *Goodman, Fanelli & Ioannidis, 2016*). Whether 'the same results' were obtained may be judged by comparing

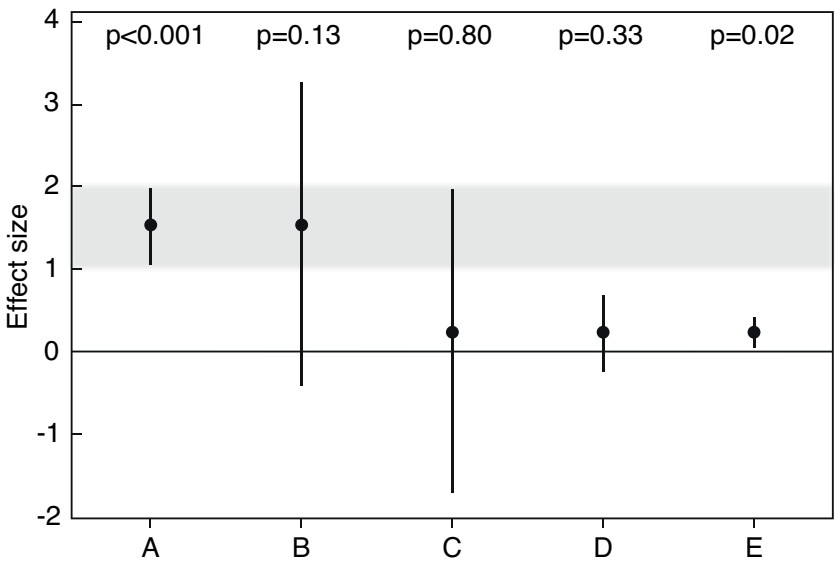

**Figure 1  Averages and 95% confidence intervals from five simulated studies.** *P*-values are from one sample *t*-tests, and sample sizes are $n = 30$ each (adapted from *Korner-Nievergelt & Hüppop, 2016*). Results A and E are relatively incompatible with the null hypothesis that the true effect size (the population average) is zero. Note that *p*-values in A versus D, or B versus C, are very different, although the estimates have the same precision and are thus equally reliable. Note also that the *p*-value in A is smaller than in E although variation is larger, because the point estimate in A is farther off the null value. If we define effect sizes between 1 and 2 as scientifically or practically important, result A is strong evidence that the effect is important, and result E is clear evidence that the effect is not important, because the small effect size was estimated with high precision. Result B is relatively clear evidence that the effect is not strongly negative and could be important, given that a value close to the center of a 95% confidence interval is about seven times as likely to be the true population parameter as is a value near a limit of the interval (*Cumming, 2014*). Result C is only very weak evidence against the null hypothesis, and because plausibility for the parameter is greatest near the point estimate, we may say that the true population average could be relatively close to zero. However, result C also shows why a large *p*-value cannot be used to 'confirm' or 'support' the null hypothesis: first, the point estimate is larger than zero, thus the null hypothesis of a zero effect is not the hypothesis most compatible with the data. Second, the confidence interval shows possible population averages that would be consistent with the data and that could be strongly negative, or positive and even practically important. Because of this large uncertainty covering qualitatively very different parameter values, we should refrain from drawing conclusions about practical consequences based on result C. In contrast, result D is only weak evidence against the null hypothesis, but precision is sufficient to infer that possible parameter values are not far off the null and that the effect is practically not important. Result C is thus a case in which the large *p*-value and the wide confidence interval roughly say the same, which is that inference is difficult. Results B and D can be meaningfully interpreted even though *p*-values are relatively large.

effect sizes and interval estimates, or by relying on subjective assessment by the scientists performing the replication (*Open Science Collaboration, 2015*). What counts in the end will be the cumulative evidential weight from multiple independent studies; and those studies will probably show considerable variation in effect sizes (*Goodman, Fanelli & Ioannidis, 2016*; *Patil, Peng & Leek, 2016*).

Traditionally, the success versus failure of a replication is defined in terms of whether an effect in the same direction as in the original study has reached statistical significance again (*Miller, 2009*; *Open Science Collaboration, 2015*; *Simonsohn, 2015*; *Fabrigar & Wegener,*

*2016*; *Nosek & Errington, 2017*). However, *p*-values are difficult to compare because they are sensitive to many differences among studies that are irrelevant to whether results are in agreement (*Greenland et al., 2016*, p. 343). For example, even if the point estimates are exactly the same, *p*-values of two studies may be on opposite sides of 0.05 because estimation precision or sample sizes differ (*Simonsohn, 2015*). Further, if the *p*-value itself is hardly replicable, we would be surprised if $p \leq 0.05$ were replicable.

So how likely will two studies both turn out to be significant? It is sometimes suggested that replicability of significance is given by the statistical power of the test used in the replication study. Power is defined as the probability that a test will be significant given that the alternative hypothesis is true. However, because in real life we do not know whether the alternative hypothesis is true, also power does not help in judging replicability of empirical results. But if we theoretically assume that the alternative hypothesis is true, we can use power to make simple calculations about the probability that *p*-values cross a significance threshold such as $p = 0.05$.

If two studies on a true alternative hypothesis have a reasonable sample size and thus the recommended statistical power of 80%, the probability that both studies are significant is 80% * 80% = 64%. As exemplified by *Greenland et al. (2016)*, this means that under the luckiest circumstances, e.g., when the alternative hypothesis is true, when there is no publication bias, and when statistical power is good, then two studies will both be significant in only 64% of cases. The probability that one study is significant and the other is not is (80% * 20%) + (20% * 80%) = 32% (*Greenland et al., 2016*; 20% is the beta error of accepting the null hypothesis although it is false, which equals 1 - power). In one third of fairly ideal replications, results will traditionally be interpreted as conflicting, and replication as having failed.

However, the above replicabilities of significance are probably overestimated for two reasons. First, replication studies often report smaller effect sizes than the original studies due to publication bias in the original studies (see below). Second, most studies end up with a power much smaller than 80%. For example, average power to detect medium sized effects was about 40–47% in 10 journals on animal behavior (*Jennions & Møller, 2003*), and median power in neuroscience was reported to be about 21% (*Button et al., 2013b*). In the Journal of Abnormal Psychology, in which *Cohen (1962)* found a median power of 46% for a medium effect in 1960, power dropped to 37% in 1984 (*Sedlmeier & Gigerenzer, 1989*). As summarized from 44 reviews on statistical power in the social, behavioral and biological sciences, average power to detect small effects was 24% (*Smaldino & McElreath, 2016*).

If we repeat the exercise by *Greenland et al. (2016)* with a more realistic power of 40%, we obtain a probability that both studies are significant of 40% * 40% = 16%, and a probability that there are conflicting results of (40% * 60%) + (60% * 40%) = 48%. This means that even if we did everything right, except for having only about average power, and if there is a true effect in the larger population, about half of our replications will fail by traditional significance standards (i.e., one study is significant and the other is not). And

only about one in six studies will significantly replicate the significant result of another study.

This is of course not the fault of the *p*-value. It is the fault of us defining replication success as the event of crossing a significance threshold. As *Greenland et al. (2016)* put it, "one could anticipate a 'replication crisis' even if there were no publication or reporting bias, simply because current design and testing conventions treat individual study results as dichotomous outputs of significant/nonsignificant or reject/accept". Even in ideal replication studies, significance as defined by classical thresholds is not to be expected, and nonsignificance cannot be used as a criterion to undermine the credibility of a preceding study (*Goodman, Fanelli & Ioannidis, 2016*).

## Significance thresholds reduce replicability

In fact, it would be highly dubious if replication success in terms of statistical significance were larger than just described. This would indicate that researchers suppress nonsignificant replications and selectively report significant outcomes, and that there is publication bias against nonsignificant studies (*Francis, 2013*). However, when nonsignificant results on a particular hypothesis remain unpublished, any significant support for the same hypothesis is rendered essentially uninterpretable (ASA statement; *Wasserstein & Lazar, 2016*). If white swans remain unpublished, reports of black swans cannot be used to infer on general swan color. In the worst case, publication bias means according to *Rosenthal (1979)* that the 95% of studies that correctly yield nonsignificant results may be vanishing in file drawers, while journals may be filled with the 5% of studies committing the alpha error by claiming to have found a significant effect when in reality the null hypothesis is true.

However, selective reporting was encouraged since the early days of significance testing. As *Fisher* (*1937*, p. 15) wrote, "it is usual and convenient for experimenters to take 5 per cent as a standard level of significance, in the sense that they are prepared to ignore all results which fail to reach this standard", an advice he gave at least since 1926 (*Fisher, 1926*). As can be read in guidelines on writing papers and theses, students should "expect that you will produce many more figures and perform many more statistical tests than will be included in the final written product" (*Lertzman, 1995*). Based on a survey of over 2,000 psychologists, *John, Loewenstein & Prelec (2012)* estimated that among researchers, the prevalence of having engaged in selective reporting of studies that 'worked' or of only a subset of measured variables is 100%.

No wonder that a bias towards publishing significant results was observed for a long time and in many research areas (*Sterling, 1959*; *Csada, James & Espie, 1996*; *Gerber & Malhotra, 2008*; *Song et al., 2010*; *Dwan et al., 2013*). Today, so-called negative results may disappear from many disciplines and countries (*Fanelli, 2012*; but see *De Winter & Dodou, 2015*), threatening the reliability of our scientific conclusions and contributing to the crisis of unreplicable research (*Gelman, 2015*). No outright fraud, no technical fault or bad experimental design are necessary to render a study irreproducible; it is sufficient that we report results preferentially if they cross a threshold of significance.

The problem with selective reporting becomes even worse because significant results are not a random sample from all possible results—significant results are biased results. If a number of studies are done on a population with a fixed effect size, studies that due

to sampling variation find a larger effect are more likely to be significant than those that happen to find smaller effects (*Schmidt, 1992*). Using statistical significance as a guideline thus typically selects for large effects that are "too good to be true" (*Gelman, 2015*). The consequence is that "most discovered true associations are inflated" (*Ioannidis, 2008*). This effect was called 'truth inflation' (*Reinhart, 2015*), or 'winner's curse' (*Zöllner & Pritchard, 2007*; *Young, Ioannidis & Al-Ubaydi, 2008*; *Button et al., 2013b*) after how the term is used in economics: in high-risk situations with competitive bidding, the winner tends to be the bidder who most strongly overestimates the value of an object being auctioned (*Capen, Clapp & Campbell, 1971*; *Foreman & Murnighan, 1996*).

The inflation of effect sizes in significant results declines as statistical power increases (for example because sample sizes are large), and inflation becomes negligible as power approaches 100% (*Colquhoun, 2014*; *Gelman & Carlin, 2014*; *Lemoine et al., 2016*). One way to see how this works is to imagine that with a power of 100%, every test becomes significant given that the alternative hypothesis is true; thus, decisions for reporting based on significance would no longer select for large effects, because every effect from every random sample would be significant. However, with a more realistic power of, e.g., 40%, only inflated effects that on average are about 1.5 times larger than the true effect size may cross the significance threshold (*Colquhoun, 2014*, Fig. 7).

In the ecological and neurological studies summarized in *Lemoine et al. (2016)* and *Button et al. (2013b)*, sample sizes required to minimize inflation of effect sizes were $n > 100$. However, as *Lemoine et al. (2016)* note, increased sample sizes can only partially offset the problem of inflated effect sizes, because a power near 100% will usually not be obtained.

Selective reporting of inflated significant effects while ignoring smaller and nonsignificant effects will lead to wrong conclusions in meta-analyses synthesizing effect sizes from a larger number of studies (*Ferguson & Heene, 2012*; *Van Assen et al., 2014*). This is one of the reasons why reliance on significance testing has been accused of systematically retarding the development of cumulative knowledge (*Schmidt, 1996*; *Branch, 2014*).

Of course, selective reporting of significant results leads to inflated effects not only in meta-analyses but in every single study. Even in cases in which authors report all conducted tests regardless of their *p*-values, but then select what to interpret and to discuss based on significance thresholds, the effects from which the authors draw their conclusions will be biased upwards.

The problem arises not only by consciously discarding nonsignificant findings. Also largely automated selection procedures may produce inflated effects, for example if genome-wide association studies select findings based on significance thresholds (*Göring, Terwilliger & Blangero, 2001*; *Garner, 2007*). In statistical model simplification, or model selection, significant predictors will have inflated point estimates (*Whittingham et al., 2006*; *Ioannidis, 2008*; *Forstmeier & Schielzeth, 2011*), and defining the importance of a predictor variable based on statistical significance will thus lead to distorted results.

## Truth inflation

The *p*-value can be seen as a measure of surprise (*Greenwald et al., 1996*): the smaller it is, the more surprising the results are if the null hypothesis is true (*Reinhart, 2015*, p. 9). If one wants to determine which patterns are unusual enough to warrant further investigation, *p*-values are thus perfectly suitable as explorative tools for selecting the largest effects. Whoever is interested in describing the average state of a system, however, should not "choose what to present based on statistical results", because "valid interpretation of those results is severely compromised" unless all tests that were done are disclosed (ASA statement, *Wasserstein & Lazar, 2016*). And, we might add, unless all results are used for interpretation and for drawing conclusions, irrespective of their *p*-values.

Of course, current incentives lead to 'significance chasing' (*Ioannidis, 2010*) rather than to publishing nonsignificant results. To put it more bluntly, "research is perverted to a hunt for statistically significant results" (*Stahel, 2016*). The road to success is nicely summarized in the online author guidelines of the journal 'Nature' (accessed 2017): "The criteria for a paper to be sent for peer-review are that the results seem novel, arresting (illuminating, unexpected or surprising)". And the *p*-value, as a measure of surprise, seems to be a great selection tool for that purpose. However, the urge for large effect sizes in novel fields with little prior research is a "perfect combination for chronic truth inflation" (*Reinhart, 2015*, p. 25). As wrote *Ioannidis (2008)*, "at the time of first postulated discovery, we usually cannot tell whether an association exists at all, let alone judge its effect size".

Indeed, the strength of evidence for a particular hypothesis usually declines over time, with replication studies presenting smaller effects than original studies (*Jennions & Møller, 2002*; *Brembs, Button & Munafo, 2013*; *Open Science Collaboration, 2015*). The reproducibility project on 100 psychological studies showed that larger original effect sizes were associated with greater effect size differences between original and replication, and that "surprising effects were less reproducible" (*Open Science Collaboration, 2015*).

Small, early, and highly cited studies tend to overestimate effects (*Fanelli, Costas & Ioannidis, 2017*). Pioneer studies with inflated effects often appear in higher-impact journals, while studies in lower-impact journals apparently tend to report more accurate estimates of effect sizes (*Ioannidis, 2005*; *Munafo, Stothart & Flint, 2009*; *Munafo & Flint, 2010*; *Siontis, Evangelou & Ioannidis, 2011*; *Brembs, Button & Munafo, 2013*). The problem is likely publication bias towards significant and inflated effects particularly in the early stages of a potential discovery. At a later time, authors of replication studies might then want, or be allowed by editors, to report results also if they found only negligible or contradictory effects, because such results find a receptive audience in a critical scientific discussion (*Jennions & Møller, 2002*). Replications therefore tend to suffer less from publication bias than original studies (*Open Science Collaboration, 2015*).

So far, academic reward mechanisms often focus on statistical significance and newsworthiness of results rather than on reproducibility (*Ioannidis et al., 2014*). Also journalists and media consumers and, therefore, all of us ask for the novel, unexpected and surprising. Thus the average truth often does not make it to the paper and the public, and much of our attention is attracted by exaggerated results.

### The earth is flat ($p > 0.05$)

The average truth might be nonsignificant and non-surprising. But this does not mean the truth equals zero. In the last decades, many authors have compiled lists with misinterpretations regarding the meaning of the $p$-value (e.g., *Greenland et al., 2016*), and surveys showed that such false beliefs are widely shared among researchers (*Oakes, 1986*; *Lecoutre, Poitevineau & Lecoutre, 2003*; *Gigerenzer, Krauss & Vitouch, 2004*; *Badenes-Ribera et al., 2016*). The ''most devastating'' of all false beliefs is probably that ''if a difference or relation is not statistically significant, then it is zero, or at least so small that it can safely be considered to be zero'' (*Schmidt, 1996*). For example, if two studies are called conflicting or inconsistent because one is significant and the other is not, it may be implicitly assumed that the nonsignificant effect size was zero (*Cumming, 2012*, p. 31).

*Cohen (1994)* published his classic critique of the use of significance tests under the title ''The earth is round ($p < .05$)''. What if this test happens to be nonsignificant? In 38%–63% of articles sampled from five journals of psychology, neuropsychology and conservation biology, nonsignificant results were interpreted as 'there is no effect', which means that a null hypothesis was accepted or 'proven' (*Finch, Cumming & Thomason, 2001*; *Schatz et al., 2005*; *Fidler et al., 2006*; *Hoekstra et al., 2006*).

In Cohen's example, 'no effect' would probably mean 'the earth is flat ($p > 0.05$).' And this is not far from reality. Similar cases abound in the published literature, such as ''lamb kill was not correlated to trapper hours ($r_{12} = 0.50$, $P = 0.095$)'' (cited in *Johnson, 1999*). It may be completely obvious that the null hypothesis of 'no effect' cannot be true, as judging from a large but nonsignificant correlation coefficient, from clear but nonsignificant differences between averages in a figure, or from common sense; but still we do not hesitate to apply our ''binary thinking, in which effects and comparisons are either treated as zero or are treated as real'' (*Gelman, 2013b*). How is this possible, since ''of course, everyone knows that one can't actually prove null hypotheses'' (*Cohen, 1990*)?

We probably tend to misinterpret $p$-values because significance testing ''does not tell us what we want to know, and we so much want to know what we want to know that, out of desperation, we nevertheless believe that it does'' (*Cohen, 1994*). Null hypothesis significance testing is not about estimating the probability that the null hypothesis or the alternative hypothesis is true—such a claim would be reserved to Bayesian testers, and even they would not be able to 'prove' any hypothesis. Null hypothesis testing is about the probability of our data *given* that the null hypothesis is true.

The problem is ''the researcher's 'Bayesian' desire for probabilities of hypotheses'' (*Gigerenzer, 1993*). It may be hopeless to temper this desire, since also Fisher himself held a ''quasi-Bayesian view that the exact level of significance somehow measures the confidence we should have that the null hypothesis is false'' (*Gigerenzer, 1993*). Yet, Fisher seemed to be clear about proving the null: a hypothesis cannot be ''proved to be true, merely because it is not contradicted by the available facts'' (*Fisher, 1935*). And, therefore, ''it is a fallacy, so well known as to be a *standard* example, to conclude from a test of significance that the null hypothesis is thereby established; at most it may be said to be confirmed or strengthened'' (*Fisher, 1955*; italics in original).

The last sentence, however, shows that also Fisher vacillated (*Gigerenzer et al., 1989*, p. 97). In fact, the null hypothesis cannot be confirmed nor strengthened, because very likely there are many better hypotheses: "Any *p*-value less than 1 implies that the test [null] hypothesis is not the hypothesis most compatible with the data, because any other hypothesis with a larger *p*-value would be even more compatible with the data" (*Greenland et al., 2016*). This can be seen when looking at a 'nonsignificant' 95% confidence interval that encompasses not only zero but also many other null hypotheses that would be compatible with the data, or, in other words, that would not be rejected using a threshold of $p = 0.05$ (Fig. 1C; *Tukey, 1991*; *Tryon, 2001*; *Hoekstra, Johnson & Kiers, 2012*). Within the confidence interval, zero is usually not the value that is closest to the observed point estimate. And even if the point estimate is exactly zero and thus "$p = 1$, there will be many other hypotheses [i.e., the values covered by the confidence interval] that are highly consistent with the data, so that a definitive conclusion of 'no association' cannot be deduced from a *p*-value, no matter how large" (*Greenland et al., 2016*).

## Limbo of suspended disbelief

It is easy to imagine research in which falsely claiming a true null effect causes great harm. To give a drastic example, *Ziliak & McCloskey* (*2008*, p. 28) cite a clinical trial on the painkiller Vioxx that reports data on heart attacks and other adverse events (*Lisse et al., 2003*). Lisse and colleagues found several '$p > 0.2$', among them that "the rofecoxib ['Vioxx'] and naproxen [generic drug] groups did not differ significantly in the number of thrombotic cardiovascular events (...) (10 vs. 7; $P > 0.2$)". The conclusion was that "the results demonstrated no difference between rofecoxib and naproxen". Later, the unjustified proof of the null caused more suffering, and the manufacturer Merck took Vioxx off the market and faced more than 4,200 lawsuits by August 20, 2005 (*Ziliak & McCloskey, 2008*).

Unfortunately, if we finally accept that we cannot accept a null hypothesis, obtaining a nonsignificant result becomes downright annoying. If the null hypothesis cannot be rejected because $p > 0.05$, it is often recommended to 'suspend judgment' (*Tryon, 2001*; *Hurlbert & Lombardi, 2009*), which leaves the null hypothesis "in a kind of limbo of suspended disbelief" (*Edwards, Lindman & Savage, 1963*). As *Cohen (1990)* put it, "all you could conclude is that you couldn't conclude that the null was false. In other words, you could hardly conclude anything".

This unfortunate state becomes even worse because usually the researchers are blamed for not having collected more data. Indeed, in correlational research, most null hypotheses of an effect of exactly zero are likely wrong at least to a small degree (*Edwards, Lindman & Savage, 1963*; *Meehl, 1967*; *Delong & Lang, 1992*; *Lecoutre & Poitevineau, 2014*—but see *Hagen, 1997*; *Hagen, 1998*; *Thompson, 1998*; *Krueger, 2001* for a critical discussion). Therefore, most tests would probably produce significant results if only one had large enough sample sizes (*Oakes, 1986*; *Cohen, 1990*; *Cohen, 1994*; *Gill, 1999*; *Stahel, 2016*). In other words, a significance test often does not make a clear statement about an effect, but instead it "examines if the sample size is large enough to detect the effect" (*Stahel, 2016*). And because "you can pretty much always get statistical significance if you look hard enough" (*Gelman, 2015*), you were probably "not trying hard enough to find significant

results" (*Ferguson & Heene, 2012*). Nonsignificance therefore seems to be regarded as "the sign of a badly conducted experiment" (*Gigerenzer et al., 1989*, p. 107).

As an outgoing editor of a major psychological journal wrote, "the [false] decision not to reject the null hypothesis can be a function of lack of power, lack of validity for the measures, unreliable measurement, lack of experiment control, and so on" (*Campbell, 1982*). Surely all of those influences could also lead to falsely claiming a significant outcome, but the editor concludes: "it is true that there is an evaluation asymmetry between significant and nonsignificant results".

Naturally, we develop an aversion to 'null results' and fail to publish them (*Ferguson & Heene, 2012*). Instead, we engage in significance chasing that may promote harmful practices like excluding variables or data with unwanted effects on *p*-values, stopping to collect data after looking at preliminary tests, rounding down *p*-values that are slightly larger than 0.05, or even falsifying data (non-exhaustive list after the survey by *John, Loewenstein & Prelec, 2012*).

Although it is now widely discussed that significant results may be much less reliable than we used to believe, mistrust in nonsignificant results still seems larger than mistrust in significant results, leading to publication bias, which in turn causes significant results to be less reliable. What could be done?

## Accepting history

First, we should briefly review the rise of modern statistics that started in the 1920s and 1930s. We think that problems like the unclear status of nonsignificant *p*-values have their roots in history, and that understanding those roots will help finding a future role for *p*-values (for more details, see *Gigerenzer et al., 1989*; *Gill, 1999*; *Salsburg, 2001*; *Lenhard, 2006*; *Hurlbert & Lombardi, 2009*; *Lehmann, 2011*).

The three most influential founding fathers, Ronald A. Fisher, Jerzy Neyman and Egon S. Pearson, disagreed on many things, but they agreed that scientific inference should not be made mechanically (*Gigerenzer & Marewski, 2015*). In their 'hypothesis tests', Neyman and Pearson confronted a point null hypothesis with a point alternative hypothesis. Based on this scenario they discovered alpha and beta errors as well as statistical power. Neyman and Pearson did not request to report *p*-values, but to make decisions based on predefined alpha and beta errors. They never recommended a fixed significance threshold (*Lehmann, 2011*, p. 55), but rather held that defining error rates "must be left to the investigator" (*Neyman & Pearson, 1933a*, p. 296).

Some years earlier, Fisher had introduced 'significance tests' using *p*-values on single null hypotheses, and he generally opposed the consideration of alternative hypotheses and of power (*Lehmann, 2011*, p. 51). Fisher did not invent the *p*-value, which he called 'level of significance', but he was the first to outline formally the logic of its use (*Goodman, 1993*). In 1925, he defined a threshold of $p = 0.05$, based on the proportion of the area under a normal distribution that falls outside of roughly two standard deviations from the mean: "The value for which $P = \cdot 05$, or 1 in 20, is 1.96 or nearly 2; it is convenient to take this point as a limit in judging whether a deviation is to be considered significant or not. Deviations exceeding twice the standard deviation are thus formally regarded as significant" (*Fisher,*

*1925*). The choice of 0.05 was acknowledged to be "arbitrary" by *Fisher* (*1929*, p. 191) and was influenced by earlier definitions of significance by William Gosset, Karl Pearson (the father of Egon) and others (discussed in *Cowles & Davis, 1982b*; *Sauley & Bedeian, 1989*; *Hurlbert & Lombardi, 2009*). A main reason why 0.05 was selected and still persists today may be that it fits our subjective feeling that events that happen at least as rarely as 10% or 1% of the time are suspiciously unlikely (*Cowles & Davis, 1982a*; *Weiss, 2011*).

Throughout his life, Fisher used the *p*-value mainly to determine whether a result was statistically significant (*Lehmann, 2011*, p. 53). In his last new book, however, Fisher famously wrote that "no scientific worker has a fixed level of significance at which from year to year, and in all circumstances, he rejects hypotheses; he rather gives his mind to each particular case in the light of his evidence and his ideas" (*Fisher, 1956*, p. 42). In the thirteenth edition of "Statistical methods for research workers" (*Fisher, 1958*), he then stated that "the actual value of *P* (...) indicates the strength of the evidence against the hypothesis" (p. 80) and that "tests of significance are used as an aid to judgment, and should not be confused with automatic acceptance tests, or 'decision functions' " (p. 128).

A main point of controversy was 'inductive inference' that was central to Fisher's thinking (*Lehmann, 2011*, p. 90). Fisher believed that significance tests allow drawing inference from observations to hypotheses, or from the sample to the population (although deducing the probability of data given that a null hypothesis is true may actually look more like *deductive* inference, from the population to the sample; *Thompson, 1999*).

In contrast, Neyman and Pearson thought that inductive inference is not possible in statistical analyses on single studies: "As far as a particular hypothesis is concerned, no test based upon a theory of probability can by itself provide any valuable evidence of the truth or falsehood of that hypothesis" (*Neyman & Pearson, 1933a*, p. 291). Their hypothesis test was therefore not meant to give a measure of evidence (*Goodman, 1993*), but to provide "rules to govern our behavior" with regard to our hypotheses, to insure that "in the long run of experience, we shall not be too often wrong" (*Neyman & Pearson, 1933a*, p. 291). The Neyman–Pearson decision procedure was particularly suitable for industrial quality control, or "sampling tests laid down in commercial specifications" (*Neyman & Pearson, 1933b*). Here, the quality of production may be tested very often over long periods of time, so that a frequentist 'long run of experience' is indeed possible, and manufacturers may indeed require thresholds to decide when to stop the production and to look for the cause of a quality problem (*Gigerenzer et al., 1989*, p. 100).

Applying Neyman–Pearson decision rules means accepting the null hypothesis while rejecting the alternative hypothesis, or vice versa. And Neyman and Pearson did indeed use the word 'accept' (e.g., *Neyman & Pearson, 1933b*). Today, we may still speak about accepting a null hypothesis when it is false as committing a beta error, or error of the second kind. But was it not said that null hypotheses cannot be accepted? Fisher wanted 'inductive inference', and for drawing scientific conclusions, a nonsignificant result can never mean to accept a null hypothesis: "errors of the second kind are committed only by those who misunderstand the nature and application of tests of significance" (*Fisher, 1935*). But Neyman and Pearson invented 'inductive behavior' explicitly for avoiding inductive inference. And for a behavioral decision, it is of course possible to accept any kind of

premises: "As Neyman emphasized, to accept a hypothesis is not to regard it as true, or to believe it. At most it means to act as if it were true" (*Gigerenzer et al., 1989*, p. 101).

Thus, accepting null hypotheses was encouraged by half of the founding schools of modern statistics. No wonder that "in approximately half of all cases, authors interpret their nonsignificant results as evidence that the null hypothesis is true" (*McCarthy, 2007*, p. 43).

Our current null hypothesis significance tests are anonymous hybrids mixing elements both from the Fisherian and the Neyman–Pearson concepts. But mixing two essentially incompatible approaches, one for measuring evidence and the other for making behavioral decisions, of course creates all sorts of problems (*Gigerenzer, 1993*; *Goodman, 1993*; *Gill, 1999*; *Hubbard & Bayarri, 2003*; *Schneider, 2015*). We often use Neyman–Pearson to refer to statistical power and to the two kinds of error. But then in practice we follow Fisher by refusing to specify a concrete point alternative hypothesis, and by interpreting exact $p$-values as graded measures of the strength of evidence against the null hypothesis (*Mundry, 2011*; *Cumming, 2012*, p. 25). However, we only consistently interpret $p$-values as strength of evidence as long as they are smaller than 0.05. For $p$-values between 0.05 and 0.1, some authors are willing to acknowledge a statistical 'trend', while others are not. For even larger $p$-values, we often switch back to a kind of Neyman–Pearson decision making, which offers no positive inference but seems to allow at least accepting the null hypothesis.

It looks like our mistrust in nonsignificant results that leads to publication bias is caused by confusion about interpretation of larger $p$-values that goes back to historical disputes among the founders of modern statistics.

## Removing significance thresholds

What could be done? We could again define what a $p$-value means: the probability of the observed data, or data more extreme, given that the null hypothesis is true. According to the interpretation by the ASA (*Wasserstein & Lazar, 2016*), smaller $p$-values cast more, and larger $p$-values cast less doubt on the null hypothesis. If we apply those definitions, it falls naturally to take $p$-values as graded evidence. We should try and forget our black-and-white thresholds (*Tukey, 1991*) and instead consider the $p$-value as a continuous measure of the compatibility between the data and the null hypothesis, "ranging from 0 for complete incompatibility to 1 for perfect compatibility" (*Greenland et al., 2016*).

We need to move away from Fisher's early recommendation to ignore nonsignificant results, because following this rule leads to publication bias and to reported effects that are biased upwards. We need to move away from the Neyman–Pearson reject/accept procedure, because it leads to proofs of the null like 'not correlated ($p = 0.095$)'. Instead, we should listen to the ASA-statisticians who say that if the "$p$-value is less than 1, some association must be present in the data, and one must look at the point estimate to determine the effect size most compatible with the data under the assumed model" (*Greenland et al., 2016*).

We are thus encouraged to interpret our point estimate as "our best bet for what we want to know" (*Cumming, 2007*). According to the central limit theorem, sample averages are approximately normally distributed, so with repeated sampling most of them would cluster around the true population average. Within a 95% confidence interval, the sample
average (the point estimate) is therefore about seven times as plausible, or seven times as good a bet for the true population average, as are the limits of the confidence interval (*Cumming, 2012*, p. 99; *Cumming, 2014*).

After looking at the point estimate, we should then interpret the upper and lower limits of the confidence interval, which indicate values that are still plausible for the true population parameter. Those values should not appear completely unrealistic or be qualitatively very different; otherwise the width of our confidence interval suggests our estimate is so noisy that we should refrain from drawing firm conclusions about practical consequences (Fig. 1C).

If necessary, we should then focus on the *p*-value as a continuous measure of compatibility (*Greenland et al., 2016*), and interpret larger *p*-values as perhaps less convincing but generally 'positive' evidence against the null hypothesis, instead of evidence that is either 'negative' or uninterpretable or that only shows we did not collect enough data. In short, we should develop a critical but positive attitude towards larger *p*-values. This alone could lead to less proofs of the null hypothesis, to less significance chasing, less data dredging, less *p*-hacking, and ultimately to less publication bias, less inflated effect sizes and more reliable research.

And removing significance thresholds is one of the smallest steps that we could imagine to address issues of replicability. Using *p*-values as graded evidence would not require a change in statistical methods. It would require a slight change in interpretation of results that would be consistent with the recommendations by the late Ronald Fisher and thus with a neoFisherian paradigm described by *Hurlbert & Lombardi (2009)*. A difference to *Hurlbert & Lombardi (2009)* is that for larger *p*-values we do not propose 'suspending judgment', which we believe would contribute to selective reporting and publication bias because we usually do not want to report results without evaluating possible conclusions. Instead, we recommend "suspending firm decisions (i.e., interpreting results with extra caution)" (*Greenland & Poole, 2013*). As we saw, some results can be meaningfully interpreted although their *p*-values are relatively large (Figs. 1B, 1D); and even if *p*-values were small, firm decisions would not be possible based on an isolated experiment (*Fisher, 1937*, p. 16).

For our next scientific enterprise using frequentist statistics, we suggest that we

(a) do our study and our analysis as planned

(b) report our point estimate, interpret our effect size

(c) report and interpret an interval estimate, e.g., a 95% confidence interval

(d) report the exact *p*-value

(e) do not use the word 'significant' and do not deny our observed effect if the *p*-value is relatively large

(f) discuss how strong we judge the evidence, and how practically important the effect is.

Do we need a scale for interpreting the strength of evidence against the null hypothesis? Graded evidence means there are no thresholds to switch from 'strong' to 'moderate' to 'weak' evidence (*Sterne & Smith, 2001*). It means that "similar data should provide similar evidence" (*Dixon, 2003*), because "surely, God loves the .06 nearly as much as the .05" (*Rosnow & Rosenthal, 1989*).

There are no 'few-sizes-fit-all' grades of evidence. Instead of following the same decision rules, no matter how large the sample size or how serious we judge measurement error, we should "move toward a greater acceptance of uncertainty and embracing of variation" (*Gelman, 2016*). If we have obtained a small *p*-value, we must be aware of the large variability of *p*-values, keep in mind that our evidence against the null hypothesis might not be as strong as it seems, and acknowledge that our point estimate is probably biased upwards. If we have obtained a large *p*-value, we must be even more aware that many hypotheses are compatible with our data, including the null hypothesis. Looking at the values covered by the confidence interval will help identifying those competing hypotheses. Very likely there are hypotheses compatible with the data that would cause even greater concern than a zero effect, e.g., if the effect would be in the opposite direction (Fig. 1C) or be much smaller or larger than what we observed in our point estimate. Note that the true effect can also be much smaller or larger than our point estimate if the *p*-value is small.

When discussing our results, we should "bring many contextual factors into play to derive scientific inferences, including the design of a study, the quality of the measurements, the external evidence for the phenomenon under study, and the validity of assumptions that underlie the data analysis" (ASA statement; *Wasserstein & Lazar, 2016*). For example, results from exploratory studies are usually less reliable than from confirmatory (replication) studies also if their *p*-values were the same, because exploratory research offers more degrees of freedom in data collection and analysis (*Gelman & Loken, 2014*; *Higginson & Munafo, 2016*; *Lew, 2016*). Already half a century ago, *Labovitz (1968)* compiled a list of criteria for evaluating evidence from *p*-values that include the practical consequences (costs) of a conclusion, the plausibility of alternative hypotheses, or the robustness of the test.

And we must keep in mind that support for our hypothesis will require not just one, but many more independent replications. If those replications do not find the same results, this is not necessarily a crisis, but a natural process by which science proceeds. In most research disciplines in which results are subject to a substantial random error, "the paradigm of accumulating evidence might be more appropriate than any binary criteria for successful or unsuccessful replication" (*Goodman, Fanelli & Ioannidis, 2016*). Thus, "the replication crisis perhaps exists only for those who do not view research through the lens of meta-analysis" (*Stanley & Spence, 2014*).

We therefore need to publish as many of our results as possible, as long as we judge them sufficiently reliable to be reported to the scientific community. And we should publish our results also if we find them neither novel nor arresting, illuminating, unexpected or surprising. Those results that we find trustworthy are the best approximates of reality, especially if they look familiar and expected and replicate what we already think we know.

## Increasing inferential reproducibility

Of course, people and journals who want to publish only small *p*-values will probably continue publishing only small *p*-values. Perhaps, "forced to focus their conclusions on continuous measures of evidence, scientists would selectively report continuous measures of evidence" (*Simonsohn, 2014*). We hope, however, that bidding farewell to significance

thresholds will free at least some of our negative results by allowing them to be positive. And we think the chances that this will happen are good: because it is already happening.

People seem prepared to interpret $p$-values as graded evidence. When asked about the degree of confidence that an experimental treatment really has an effect, the average researcher's confidence was found to drop quite sharply as $p$-values rose to about 0.1, but then confidence levelled off until 0.9, essentially showing a graded response (*Poitevineau & Lecoutre, 2001*). *Nuzzo (2015)* cites Matthew Hankins, who has collected "more than 500 creative phrases that researchers use to convince readers that their nonsignificant results are worthy of attention (see go.nature.com/pwctoq). These include 'flirting with conventional levels of significance ($p > 0.1$)', 'on the very fringes of significance ($p = 0.099$)' and 'not absolutely significant but very probably so ($p > 0.05$)'."

As *Hurlbert & Lombardi (2009)* note, even *Neyman* (*1977*, p. 112) labelled $p$-values of 0.09, 0.03, and <0.01 reported in an earlier paper (*Lovasich et al., 1971*, Table 1, right column) as 'approximately significant', 'significant', and 'highly significant', respectively. And creativity is increasing. *Pritschet, Powell & Horne (2016)* searched over 1,500 papers in three journals of psychology for terms like 'marginally significant' or 'approaching significance' that were used for $p$-values between 0.05 to 0.1 and up to 0.18. They found that "the odds of an article describing a $p$-value as marginally significant in 2010 were 3.6 times those of an article published in 1970." In 2000 and 2010, the proportions of articles describing at least one $p$-value as marginally significant were 59% and 54%.

The rules how to label results if $p > 0.05$ are usually unwritten (*Pritschet, Powell & Horne, 2016*), so it is current practice to interpret $p$-values between 0.05 and 0.1, or even larger $p$-values, as evidence either against the null hypothesis, or (falsely) in favor of a null effect, or as no evidence at all. This anarchical state of affairs undermines 'inferential reproducibility', which might be the most important dimension of reproducibility and "refers to the drawing of *qualitatively* similar conclusions from either an independent replication of a study or a reanalysis of the original study" (*Goodman, Fanelli & Ioannidis, 2016*; italics supplied). Interpreting larger $p$-values as graded evidence, but as evidence that can only speak against the null hypothesis, would clearly help increasing inferential reproducibility by reducing the choices for qualitative interpretation. For example, a large $p$-value would not be taken to support the null hypothesis, and only rarely would a result be interpreted as providing no evidence at all.

As listed in the beginning of this paper, many authors have argued for removing fixed thresholds. In 2016, *Lemoine et al. (2016)* wrote: "we join others in calling for a shift in the statistical paradigm away from thresholds for statistical significance", and similar words were used in the closing remarks of *Greenland et al. (2016)*. So far, however, we see few researchers performing this shift. Although black-and-white seems to give way to a more flexible approach allowing for trends and marginal significance, dichotomous threshold thinking is still the rule among applied researchers and even among many statisticians (*McShane & Gal, 2016*; *McShane & Gal, in press*). Perhaps this is because of serious issues with interpreting $p$-values as graded evidence?

## Arguments against removing thresholds

In the following, we discuss potential problems. We start each paragraph with a possible argument that could be raised against removing fixed significance thresholds.

*'We need more stringent decision rules'*

Evidently, removing significance thresholds "may lead to an increased prevalence of findings that provide weak evidence, at best, against the null hypothesis" (*Pritschet, Powell & Horne, 2016*). This will not cause any problem, as long as the authors of a study acknowledge that they found only weak evidence. Quite the contrary, publishing weak evidence is necessary to reduce publication bias and truth inflation. However, $p$-values just below 0.05 are currently interpreted as generally allowing a decision to reject the null hypothesis, which is one of the reasons why scientific claims may often be unreliable (*Oakes, 1986*; *Sterne & Smith, 2001*; *Johnson, 2013*; *Colquhoun, 2014*). One alternative proposition to enhance replicability of research was therefore not to remove thresholds, but rather to apply more stringent thresholds (*Johnson, 2013*; *Ioannidis, 2014*; *Academy of Medical Sciences, 2015*). For example, it may be said that you should "not regard anything greater than $p < 0.001$ as a demonstration that you have discovered something" (*Colquhoun, 2014*).

We agree it is currently too "easy for researchers to find large and statistically significant effects that could arise from noise alone" (*Gelman & Carlin, 2014*). Interpreting $p$-values as graded evidence would mean to mistrust $p$-values around 0.05, or smaller, depending on the circumstances and the scientific discipline. To announce a discovery in particle physics, a $p$-value as small as 0.0000003 may be needed (*Johnson, 2014*). Also in other disciplines, we should often judge our evidence as strong only if the $p$-value is much smaller than 0.05: if we want to demonstrate a surprising, counterintuitive effect; if we know that our null hypothesis has a high prior probability (*Bayarri et al., 2016*); if our sample size is large (*Anderson, Burnham & Thompson, 2000*; *Pericchi, Pereira & Perez, 2014*); or if postulating an effect that in reality is negligible would have serious practical consequences.

However, even a $p$-value of 0.1 or larger may be interpreted as sufficiently strong evidence: if we collected our data truly randomized and by minimizing bias and measurement error; if we stuck to our pre-planned protocol for data analysis without trying multiple alternative analyses; if our effect size is small; or if claiming only weak evidence for an effect that in reality is practically important would have serious consequences (*Gaudart et al., 2014*).

A large effect with a large $p$-value could potentially have much more impact in the real world than a small effect with a small $p$-value (*Lemoine et al., 2016*), although in the first case, the evidence against the null hypothesis is weaker (Figs. 1B, 1E). And in exploratory studies screening data for possible effects, results may be considered interesting even if their $p$-values are large (*Madden, Shah & Esker, 2015*). Since $p$-values from exploratory research should be taken as descriptive with little inferential content (*Berry, 2016*), such studies are perhaps the clearest cases in which there is no need for significance thresholds (*Lew, 2016*). When exploring "what the data say" (*Lew, 2016*), it simply makes no sense to listen to them only if $p < 0.05$. We should, however, clearly state that the research was exploratory (*Simmons, Nelson & Simonsohn, 2011*; *Berry, 2016*). And we should keep in mind that if

we recommend effects for further investigation because their *p*-values are small, our effect sizes are likely inflated and will probably be smaller in a follow-up study.

Another argument against stringent thresholds like $p = 0.001$ is that sample sizes in fields like biomedicine or animal behavior are often bound to be small for practical and ethical reasons (*Nakagawa, 2004*; *Gaudart et al., 2014*; *Academy of Medical Sciences, 2015*). With small or moderate sample sizes, the chances that small but important effects will not become significant is very high, and this applies both to more stringent thresholds and to a possible scenario in which the conventional threshold of 0.05 were reinforced.

Because smaller *p*-values usually come with larger sample sizes, the question whether we should aim for smaller *p*-values boils down to whether we should conduct fewer but larger or more but smaller studies. Confidence and precision clearly increase with sample size (*Button et al., 2013a*). Based on simulations, however, *IntHout, Ioannidis & Borm (2016)* recommend doing rather several studies with a moderate power of 30%–50% than one larger study that would need a sample size four times to twice as large, respectively, to obtain 80% power. The reason is that every study will suffer from some sort of bias in the way it is conducted or reported, and that "in a series of studies, it is less likely that all studies will suffer from the same type of bias; consequently, their composite picture may be more informative than the result of a single large trial" (*IntHout, Ioannidis & Borm, 2016*).

And of course, to be of any use, all of those studies should be published. Cumulative evidence often builds up from several studies with larger *p*-values that only when combined show clear evidence against the null hypothesis (*Greenland et al., 2016*, p. 343). Very possibly, more stringent thresholds would lead to even more results being left unpublished, enhancing publication bias (*Gaudart et al., 2014*; *Gelman & Robert, 2014*). What we call winner's curse, truth inflation or inflated effect sizes will become even more severe with more stringent thresholds (*Button et al., 2013b*). And one reason why 'false-positives' are vastly more likely than we think, or as we prefer to say, why the evidence is often vastly overestimated, is that we often flexibly try different approaches to analysis (*Simmons, Nelson & Simonsohn, 2011*; *Gelman & Loken, 2014*). Exploiting those researcher degrees of freedom, and related phenomena termed multiplicity, data dredging, or *p*-hacking, would probably become more severe if obtaining significant results were harder due to more stringent thresholds.

We think that while aiming at making our published claims more reliable, requesting more stringent fixed thresholds would achieve quite the opposite.

*'Sample sizes will decrease'*

Ideally, significance thresholds should force researchers to think about the sizes of their samples—this is perhaps the only advantage thresholds could potentially offer. If significance is no longer required, it could happen that researchers more often content themselves with smaller sample sizes.

However, the argument could as well be reversed: fixed significance thresholds may lead to unreasonably small sample sizes. We have had significance thresholds for decades, and average power of studies was constantly low (*Sedlmeier & Gigerenzer, 1989*; *Button et al., 2013b*). Researchers seem often to rely on rules of thumb by selecting sample sizes they

think will yield *p*-values just below 0.05, although it is clear that *p*-values in this order of magnitude are hardly replicable (*Vankov, Bowers & Munafo, 2014*). *Oakes* (*1986*, p. 85–86) argues that the major reason for the abundance of low-power studies is the belief that "experimental findings suddenly assume the mantle of reality at the arbitrary 0.05 level of significance"—if a result becomes true once a threshold is crossed, there is simply no need to strive for larger sample sizes.

In their summary of 44 reviews on statistical power, *Smaldino & McElreath (2016)* found not only that average power to detect small effects was 24%, but that there was no sign of increase over six decades. The authors blame an academic environment that only rewards significant findings, because in such a system, "an efficient way to succeed is to conduct low power studies. Why? Such studies are cheap and can be farmed for significant results" (*Smaldino & McElreath, 2016*). Similarly, *Higginson & Munafo (2016)* suggest that "researchers acting to maximize their fitness should spend most of their effort seeking novel results and conduct small studies that have only 10%–40% statistical power".

Quite often, significance testing appears like a sort of gambling. Even a study with minimized investment into sample sizes will yield significant results, if only enough variables are measured and the right buttons in the statistical software are pushed. Small sample sizes further have it that significant effects will probably be inflated, so large and surprising effects are almost guaranteed. And we may have become somewhat addicted to this game—it is satisfying to feed data into the machine and find out whether the right variables have turned significant.

Indeed, "Fisher offered the idea of *p*-values as a means of protecting researchers from declaring truth based on patterns in noise. In an ironic twist, *p*-values are now often manipulated to lend credence to noisy claims based on small samples" (*Gelman & Loken, 2014*). And the manipulation can happen completely unintentionally, "without the researcher performing any conscious procedure of fishing through the data" (*Gelman & Loken, 2014*), just as a side-effect of how the game of significance testing is usually applied.

We should discourage significance farming by insisting that a *p*-value of 0.02 is not automatically convincing, and that there is no need to throw away a study with $p = 0.2$. We hope that removing significance thresholds will allow researchers to take the risk and put more effort into larger studies, without the fear that all the effort could be wasted if there were nonsignificance in the end.

*'We need objective decisions'*

Graded evidence means no strict criterion to decide whether there is evidence or not, no algorithm that makes the decision for us. Can graded evidence be objectively interpreted? Fisher's honorable aim was to develop a "rigorous and objective test" (*Fisher, 1922*, p. 314), and Neyman and Pearson made the test even more rigorous by introducing automatic decisions between two competing hypotheses.

Unfortunately, it seems like "the original Neyman–Pearson framework has no utility outside quality control type applications" (*Hurlbert & Lombardi, 2009*). In most studies, we do not formally specify alternative hypotheses, and we tend to use small sample sizes, so that beta error rates are usually high and unknown (*Fidler et al.,*

*2004*). This is one reason why our Fisher–Neyman–Pearson hybrid tests offer only an "illusion of objectivity" (*Berger & Berry, 1988*; *Gigerenzer, 1993*). Another reason is that all statistical methods require subjective choices (*Gelman & Hennig, in press*). Prior to calculating $p$-values, we make all kinds of personal decisions: in formulating our research question, in selecting the variables to be measured, in determining the data sampling scheme, the statistical model, the test statistic, how to verify whether model assumptions are met, how to handle outliers, how to transform the data, which software to use. We do all of that and are used to justifying our choices; but interestingly, when it comes to interpreting test results, "one can feel widespread anxiety surrounding the exercise of informed personal judgment" (*Gigerenzer, 1993*).

For statistical tests, our *a priori* assumptions about the model may be, for example, that residuals are independent and identically distributed. Since "it is impossible logically to distinguish between model assumptions and the prior distribution of the parameter", using prior information is not a feature peculiar to Bayesian inference, but a necessity for all scientific inference (*Box, 1980*). As explained by *Oakes* (*1986*, p. 114), the statistical schools differ in the manner in which they employ this prior information.

*Pearson (1962)* wrote about his work with Neyman: "We were certainly aware that inferences must make use of prior information", thus "we left in our mathematical model a gap for the exercise of a more intuitive process of personal judgment in such matters—to use our terminology—as the choice of the most likely class of admissible hypotheses, the appropriate significance level, the magnitude of worthwhile effects and the balance of utilities." But these judgments had to be made before data collection, and "every detail of the design, including responses to all possible surprises in the incoming data, must be planned in advance" (*Berger & Berry, 1988*). To change the plan after data collection is to violate the model (*Oakes, 1986*).

We agree this is the way to go in confirmatory studies that replicate procedures from earlier research. Sticking to the rules is one reason why results from replication studies are usually more reliable than from exploratory studies (*Gelman & Loken, 2014*; *Higginson & Munafo, 2016*); another reason is that confirmatory studies may suffer less from publication bias than exploratory studies (*Open Science Collaboration, 2015*). We remind, however, that even in ideal confirmatory studies, significance is not to be expected (see above), thus significance thresholds should not be applied to judge replication success. And in exploratory studies, we see even less utility for the original Neyman–Pearson framework and their predefined significance thresholds. Such studies have all the rights to interrogate the data repeatedly and intensively (*Lew, 2016*), as long as they are acknowledged to be exploratory (*Gelman & Loken, 2014*; *Berry, 2016*), and, of course, as long as they report all results irrespective of their $p$-values.

We thus side with Ronald Fisher that the decision how to deal with the null hypothesis should reside with the investigator rather than be taken for him in the Neyman–Pearson manner (*Oakes, 1986*, p. 114). It should be remembered, however, that the decision whether there is an important effect cannot be answered in a single study, and that it is often more interesting to discuss the size of the effect than to speculate on its mere existence (e.g., *Cumming, 2012*; *Gelman, 2013a*). In many cases, "research requires no firm decision: it

contributes incrementally to an existing body of knowledge" (*Sterne & Smith, 2001*). Or in the words of *Rozeboom (1960)*: "The primary aim of a scientific experiment is not to precipitate decisions, but to make an appropriate adjustment in the degree to which one accepts, or believes, the hypothesis or hypotheses being tested".

*'Null hypotheses should be acceptable'*

As stated earlier, most null hypotheses of an effect of exactly zero are likely wrong at least to a small degree. This claim seems to be widely accepted for correlational research (*Edwards, Lindman & Savage, 1963*; *Meehl, 1967*; *Cohen, 1990*; *Tukey, 1991*; *Cohen, 1994*; *Lecoutre & Poitevineau, 2014*; *Stahel, 2016*), although its general applicability and philosophical background are debated (*Hagen, 1997*; *Hagen, 1998*; *Thompson, 1998*; *Krueger, 2001*). However, the claim may not apply to experimental studies (*Meehl, 1990*). A null hypothesis of an effect of exactly zero may be valid "for true experiments involving randomization (e.g., controlled clinical trials) or when any departure from pure chance is meaningful (as in laboratory experiments on clairvoyance)" (*Cohen, 1994*).

Under such circumstances, we would like to be able to show that the null hypothesis is true. But how should this be done, if any $p$-value that we calculate can only speak against the null hypothesis (*Delong & Lang, 1992*)? Interpreting $p$-values as graded evidence will mean that we can never support the null hypothesis, and that alternative hypotheses may be "virtually unkillable", buried at "a vast graveyard of undead theories" (*Ferguson & Heene, 2012*, although the authors coined those words decrying publication bias against 'null' results and not the use of graded evidence).

So "how to accept the null hypothesis gracefully" (*Greenwald, 1975*)? The hard truth is that $p$-values from classical null hypothesis tests are not made for this task—whether or not we use them as graded evidence. As we discussed at length, also nonsignificant results cannot support the null hypothesis, and high power is of no help (*Greenland, 2012*). There is only one way to attach a number to the probability of a null hypothesis: "use a range, rather than a point, null hypothesis", and "compute the posterior probability of the null (range) hypothesis" (*Greenwald, 1975*), or use similar Bayesian methods (*Gallistel, 2009*; *Morey & Rouder, 2011*; *Dienes, 2014*). Alternative procedures in a frequentist framework include equivalence tests (*Levine et al., 2008*) or the careful interpretation of effect sizes and confidence intervals, to determine whether an effect is so precisely small as to be practically unimportant (Figs. 1D, 1E; *Cumming, 2014*, p. 17; *Simonsohn, 2015*).

*'We need to get rid of p-values'*

So $p$-values vary enormously from sample to sample even if there is a true effect. Unless they are very small, $p$-values therefore give only unreliable evidence against the null hypothesis (*Cumming, 2012*, p. 152). For this and other reasons, $p$-values should probably be viewed as descriptive statistics, not as formal quantifications of evidence (*Lavine, 2014*). To make it worse, we usually translate the observed evidence against the null hypothesis into evidence in favor of an alternative hypothesis, without even mentioning the null hypothesis ('females were twice as large as males, $p < 0.001$'). However, knowing that the data are unlikely under the null hypothesis is of little use unless we consider whether or

not they are also unlikely under the alternative hypothesis (*Sellke, Bayarri & Berger, 2001*; *Barber & Ogle, 2014*). Since very often we are not honestly interested in describing evidence *against* a hypothesis, null hypothesis testing is usually a "parody of falsificationism in which straw-man null hypothesis A is rejected and this is taken as evidence in favor of preferred alternative B" (*Gelman, 2016*).

Why do we not join others and discourage using *p*-values and null hypothesis tests (e.g., *Carver, 1978*; *Schmidt, 1996*; *Ziliak & McCloskey, 2008*; *Orlitzky, 2012*; *Cumming, 2014*; *Trafimow & Marks, 2015*; *Gorard, 2016*)?

We agree to *Cumming* (*2012*, p. 33) that "thinking of *p* as strength of evidence may be the least bad approach". We thus propose to describe our observed gradual evidence against the null hypothesis rather than to 'reject' the null. A first step would be to stop using the word 'significant' (*Higgs, 2013*; *Colquhoun, 2014*). Indeed, often null hypotheses about zero effects are automatically chosen only to be rejected (*Gelman, 2013b*), and null hypotheses on effect sizes other than zero, or on ranges of effect sizes, would be more appropriate (*Cohen, 1994*; *Greenland et al., 2016*). The way to become aware of our zero effect automatism is to "replace the word significant with a concise and defensible description of what we actually mean to evoke by using it" (*Higgs, 2013*).

These are small changes that may look insufficient to some. But the beauty in such simple measures to help address the crisis of unreplicable research is that they do not hurt. We do not need to publicly register our study protocol before we collect data, we do not need to learn new statistics, we do not even need to use confidence intervals if we prefer to use standard errors. Of course all of those measures would be helpful (e.g., *Cumming, 2014*; *Academy of Medical Sciences, 2015*), but they usually require an active change of research practices. And many researchers seem to hesitate changing statistical procedures that have been standard for nearly a century (*Thompson, 1999*; *Sharpe, 2013*).

Although there are hundreds of papers arguing against null hypothesis significance testing, we see more and more *p*-values (*Chavalarias et al., 2016*) and the ASA feeling obliged to tell us how to use them properly (*Goodman, 2016*; *Wasserstein & Lazar, 2016*). Apparently, bashing or banning *p*-values does not work. We need a smaller incremental step that at the same time is highly efficient. There are not many easy ways to improve scientific inference, but together with *Higgs (2013)* and others, we believe that removing significance thresholds is one of them.

Also, we do not deny that some thresholds are necessary when interpreting statistics. If we want to draw error bars around a mean, the lines of the bars must end at some point that is defined by a threshold. By convention, we use 95% confidence intervals, cutting off parameter values that would be rejected at the threshold of $p = 0.05$; of course we could also use 90%, 80% or 75% confidence intervals with thresholds of 0.10, 0.20 or 0.25 (*Hurlbert & Lombardi, 2009*). *Cumming*'s (*2012*, p. 76) recommendation is to concentrate on 95% confidence intervals, because "it's challenging enough to build up good intuitions about the standard 95% level of confidence".

Unfortunately, many researchers seem to use confidence intervals mostly to decide whether the null value is excluded, thus converting them to significance tests (*Hoekstra, Johnson & Kiers, 2012*; *McCormack, Vandermeer & Allan, 2013*; *Rothman, 2014*; *Savalei &*

*Dunn, 2015*; *Van Helden, 2016*). Potentially, using confidence intervals could encourage estimation thinking and meta-analytic thinking (*Cumming, 2014*), but for full benefit, researchers would need to interpret confidence intervals without recourse to significance (*Coulson et al., 2010*). 'Cat's-eye' pictures of 95% confidence intervals show how the plausibility that a value is the true population average is greatest for values near our point estimate, in the center of the interval; plausibility then drops smoothly to either end of the confidence interval, then continues to drop further outside the interval (*Cumming, 2012*, p. 95). This means that "we should not lapse back into dichotomous thinking by attaching any particular importance to whether a value of interest lies just inside or just outside our confidence interval" (*Cumming, 2014*).

Further, we recommend choosing not only from the available null hypothesis tests but also from the toolbox provided by Bayesian statistics (*Korner-Nievergelt et al., 2015*). But we agree that we should not "look for a magic alternative to NHST [null hypothesis significance testing], some other objective mechanical ritual to replace it. It doesn't exist" (*Cohen, 1994*).

Whatever method of statistical inference we use, biased effect sizes will be a problem with criteria that are used to select results for publication or interpretation. If effect sizes are reported because they are large, this will of course create an upwards bias, similarly to selecting $p$-values because they are small. If effect sizes are reported because they are small, or $p$-values because they are large, for example when people are interested to show that some treatment may not have an (adverse) effect, this will create a downwards bias (*Greenland et al., 2016*). If results are reported because Bayesian posterior probabilities are at least three times larger for an alternative than for a null hypothesis, this is equivalent to selective reporting based on significance thresholds (*Simonsohn, 2014*).

Inflated effects will occur when a discovery is claimed because a Bayes factor is better than a given value or a false discovery rate is below a given value (*Button et al., 2013b*). Selecting a model because the difference in the Akaike information criterion ($\Delta$AIC) passes some threshold is equivalent to model selection based on significance and will generate inflated effects (*Murtaugh, 2014a*; *Murtaugh, 2014b*; *Parker et al., 2016*). Whenever we use effect sizes, confidence intervals, AIC, posteriors, Bayes factors, likelihood ratios, or false discovery rates in threshold tests and decision heuristics for reporting or interpreting selected results rather than all results, we create biases and invalidate the answers we give to our questions (*Simonsohn, 2014*; *Yu et al., 2014*; *Parker et al., 2016*; *Wasserstein & Lazar, 2016*).

However, the greatest source of bias probably comes from selectively reporting small $p$-values, simply because of the dominant role of null hypothesis significance testing. If we learn to judge the strength of evidence based not on the event of passing a threshold, but on graded summaries of data like the $p$-value, we will become more aware of uncertainty. And statistical methods are not simply applied to a discipline but change the discipline itself (*Gigerenzer & Marewski, 2015*). A greater acceptance of uncertainty and embracing of variation (*Gelman, 2016*) could shift our focus back to core values like discussing practical importance of effect sizes, minimizing measurement error, performing replication studies, and using informed personal judgment (*Cumming, 2014*; *Gigerenzer & Marewski, 2015*; *Lemoine et al., 2016*).

## CONCLUSIONS

Part of an apparent crisis of unreplicable research is caused by the way we use statistics for analyzing, interpreting and communicating our data. Applying significance thresholds leads to overconfidence in small but highly variable $p$-values, to conclusions that are based on inflated effects, and to publication bias against larger $p$-values. But larger $p$-values are to be expected also if there is a true effect, and they must be published because otherwise smaller $p$-values are uninterpretable. Thus, smaller $p$-values need to lose reputation, and larger $p$-values need to gain reputation. This is best accomplished by removing fixed significance thresholds, by cautiously interpreting $p$-values as graded evidence against the null hypothesis, and by putting more emphasis on interpreting effect sizes and interval estimates, using non-automated informed judgment. We give the last word to *Boring (1919)*, who one century ago wrote in his paper "Mathematical vs. scientific significance": "Conclusions must ultimately be left to the scientific intuition of the experimenter and his public."

## ACKNOWLEDGEMENTS

For helpful comments on the manuscript we thank Daniel Berner, Lilla Lovász, the students and colleagues from our journal clubs, and the three referees.

### Funding

Financial support was provided by the Swiss National Science Foundation (grant no. 156294), the Swiss Association Pro Petite Camargue Alsacienne, the Fondation de bienfaisance Jeanne Lovioz and the MAVA Foundation. The funders had no role in study design, data collection and analysis, decision to publish, or preparation of the manuscript.

### Grant Disclosures

The following grant information was disclosed by the authors:
Swiss National Science Foundation: 156294.
Swiss Association Pro Petite Camargue Alsacienne.
Fondation de bienfaisance Jeanne Lovioz.
MAVA Foundation.

### Competing Interests

The authors declare there are no competing interests.

### Author Contributions

- Valentin Amrhein conceived and designed the experiments, wrote the paper, prepared figures and/or tables, reviewed drafts of the paper.
- Fränzi Korner-Nievergelt conceived and designed the experiments, reviewed drafts of the paper.
- Tobias Roth conceived and designed the experiments, reviewed drafts of the paper.

## Data Availability

This is a review without empirical data.

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
