# Peer review of "The earth is flat (p > 0.05): significance thresholds and the crisis of unreplicable research"

_PeerJ, doi:10.7717/peerj.3544_

## Round 0.1 · original submission · Major Revisions

While Reviewer 1 and 3 found your review meaningful and informative, Reviewer 2 has substantial concerns about the novelty of your review and the viewpoint you support, i.e., the neoFisherian interpretation. The writing of the manuscript could be more concise, precise and structured. Please pay special attention to the comments of Reviewer 2 in the rebuttal letter.

·

Basic reporting

This article is written in good, but not excellent, English. The authors have an engaging style of writing, but I would rather at times this was slightly more formal and less wordy. I would also urge the authors to provide definitions to terms with which not everyone is familiar, e.g. data dredging, p-hacking, publication bias, etc.

The citations within the article are excellent and cover most classical papers in the field.

Experimental design

I think that the abstract of this work would derive much benefit from including the following within the abstract: aims and objectives of their review, its perceived significance, innovativeness and its conclusions.

Validity of the findings

Conclusions more or less follow from the discussion. I have made certain comments below.

Additional comments

Line 71: I am not aware of Boring (1919), but modern hypothesis testing emerged in the 1920’s after work by Fisher and Neyman-Pearson.

Line 78-79: Very good point. You may want to cite Fisher’s work (which is also beautifully described in Steve Goodman’s paper) on p-values, his objection to using them as a mechanistic threshold and his concern that Neyman-Pearson’s approach would lead to that. I see that you cited this paper in 532, excellent.

Line 89: Your reference is excellent – please point out that this is a news feature, not a peer-reviewed research article.

Line 152: This is not entirely true. For example, hypothesis testing for measures of relative risk (e.g. odds ratios) takes relative risk = 1 as the null (unless you consider those risks in the exponential scale).

Line 165: I think that this is an excellent time to mention Ioannidis’s paper on small study effects.

Line 173: I would be more careful with mentioning uniform distributions of p-values. Some statisticians have provided counterarguments to the notion of uniform distribution, even though it is broadly true.

Line 184-186: You may want to mention Ioannidis’s ‘Proteus phenomenon’ here.

Line 192: Please define what you mean by prediction intervals vs confidence intervals. Most readers would not know the difference.

Line 205-206: p-values are calculated using values that asymptotically approximate population values, as per the central limit theorem. The sample average has an uncertainty described by the standard error of the mean, but its value is asymptotically identical to the population mean.

Line 216: what do you mean by using p-values as a statistic? By ‘statistic’ we refer to attributes of a population, such as the mean age of a population. P-values are not a statistic.

Line 300-305: You may want to cite Ioannidis’s paper on Why most research findings are false here.

Line 350-353: Most meta-analyses are not averaging results across studies, unless very specific conditions apply (e.g. absence of heterogeneity, which is rare). Please replace ‘averaging’ with ‘synthesising’. Also, in doing a meta-analysis researchers use a variety of measures to identify and quantify selective reporting.

Line 360-366: Indeed, using automated selection procedures inappropriately is often wrong, especially when our aim is inference and not prediction. However, appropriate use of automated selection procedures in the right context and in combination with certain validation techniques, such as cross-validation, is far from wrong.

Line 955-962: I do not think it is prudent to dismiss those proposals because they address issues that simply using p-values appropriately does not resolve. For example, pre-registered study protocols aim to address the issue of selective reporting of statistical analyses. Furthermore, relatively new statistical techniques have identified ways of obtaining more reliable results, especially in relation to prediction (e.g. lasso, tree-based methods, etc.) and observational studies (e.g. propensity scores, instrumental variables). I would change the wording of this paragraph.

Line 964-965: I cannot see how these papers have led to more use of p-values, nor have the authors provided adequate evidence to suggest such a causal link. I would reword this.

Line 995-997: I agree with the final sentence of this paragraph. All results should be reported openly and transparently and all datasets should be made available upon publication (not upon request to the authors). However, there are parts of the paragraph prior to that last sentence, which I found confusing and better omitted or at the very least reworded. These relate to the use of Bayes’ factors, AIC and other measures. I will also take this opportunity to remind the authors of another very important technique, that of false discovery rates; I believe that the authors should also mention this when referring to possible alternatives to p-values.

###

Is the review of broad and cross-disciplinary interest and within the scope of the journal? Yes

Has the field been reviewed recently? If so, is there a good reason for this review (different point of view, accessible to a different audience, etc.)? This field has been reviewed multiple times in a much more eloquent manner and I have seen all of the ideas contained in this review before. Nevertheless, it provides a very good resource of classical papers in the field, for which reason readers unaware of this literature may derive benefit from it, even though this article often uses jargon that could deter such audience.

Does the Introduction adequately introduce the subject and make it clear who the audience is/what the motivation is? The introduction is not structured appropriately and it is difficult to know the motivation of the authors as well as what is to follow on the basis of the abstract/introduction.

Is the Survey Methodology consistent with a comprehensive, unbiased coverage of the subject. If not, what is missing? The authors have provided a very good account of the literature in the field. A very important method that they have not mentioned is false discovery rate, which is though more relevant to large datasets.

Are sources adequately cited? Quoted or paraphrased as appropriate. Yes.

Is the review organized logically into coherent paragraphs/subsections? I believe that it could be much better structured and much less wordy.

Is there a well-developed and supported argument that meets the goals set out in the Introduction? No clear goals were set in the introduction. The evidence that the authors provide support some, but not all of their conclusions (please see comments to the authors above).

Does the Conclusion identify unresolved questions / gaps / future directions? It identifies advice to researchers, which is good and valid, even though stated repeatedly previously.

·

Basic reporting

There are hundreds of articles like this manuscript, pointing out errors in the way people use statistics. One of the peer review guidelines at peerJ ask me: “Has the field been reviewed recently? If so, is there a good reason for this review (different point of view, accessible to a different audience, etc.)?” And I have to conclude this topic has been reviewed literally hundreds of times, and the current article adds nothing. Moreover, many extremely important topics are not addressed, and the viewpoint the authors support (a neoFisherian interpretation) is so severely flawed, it would make statistical inferences people make even worse if it was adopted.

This article jumps around in a very unstructured manner, briefly discussing a wide range of issues, from overestimation of effect sizes to the history of NHST, without going in depth, clarity, or novelty on any of these issues. The article really starts at line 605, and everything before then can be cut without any loss, and probably a lot of improvement, in coherence. The authors repeat previous recommendation to use p-values as graded evidence, without acknowledging well-known problems with this (e.g., Lindley’s paradox). The neo-Fisherian approach the authors follow is severely flawed, but the authors do not discuss these flaws (i.e., p-values are at best correlated with evidence, but can never be used as evidence, especially not when the alternative is not quantified). The authors fail to discuss real improvements (e.g., to deal with non-significant results, we need to use equivalence testing in addition to testing the null-hypothesis). If you want to report evidence, report likelihoods or Bayesian statistics, not p-values. The authors don’t even discuss these approaches.

I don’t like articles that conflate the strengths and weaknesses of p-values, with reproducibility, and p-values with NHST. These issues are completely orthogonal. Well-applied Neyman-Pearson statistics is as likely to lead to scientific progress as other approaches. Reproducibility problem come from designing bad studies, having bad theories, bad incentives, and publication bias.

You can not ‘remove’ a practice, without putting something else in its place, just like you can not abandon a theory, without putting something in its place. And the authors really put nothing detail in place of current practices, that has not been mentioned at least a thousand times. This lack of novelty is not a problem is it would be accompanied by great clarity, but this is not the case. I would urge the authors to carefully re-read the article. It is not written well. The article does not flow, but it reads like a list of points with very little structure. Many sections contain repetitive sentences, and I think the same point can be made in 50% of the words. The authors discuss many things that, they themselves acknowledge, have little to do with p-values. It reads like a strong of quotes and summaries of papers written on this topic. The first time I read the article, I did not uncover what they point was that the authors were trying to make, except reviewing the literature in a course and unstructured manner.

Experimental design

NA

Validity of the findings

NA

Additional comments

I’m sorry I can not be more positive. I appreciate the authors have thought carefully about these issues, and read a lot of papers. But I don’t see any clear contribution to the literature in this manuscript.

Reviewer 3 ·

Basic reporting

The manuscript is well-written and well-formatted. It includes an impressive number of (relevant) references. Professional English is used throughout.

Experimental design

Only partially relevant here (the manuscript is an essay).
But yes, the topic is well-defined, relevant and meaningful.

Validity of the findings

Not really relevant here (no experimental results). But yes, the paper seems scientifically valid to me.

Additional comments

This manuscript is an essay on the interpretation of p-values. It handles a very important topic, which is currently attracting a lot of attention in the literature (see the initiative by the American Statistical Association). The manuscript is well-written and gives an excellent up-to-date overview of the literature in this field. It also suggests own valuable ideas. It was a pleasure to read this manuscript. I believe that such papers on the philosophical foundations of scientific discovery deserve their place in journals read not (just) by philosophers but (also) by scientists using the considered methods for their scientific discoveries.

Please find below a list of comments that should not be very difficult to address and might help to improve the manuscript.

Let me also mention three very recent papers that may give inspiring feed for thought to the authors (with respect to the present manuscript, especially the section "We need objective decisions", or for their later manuscripts):
- Gelman & Hennig's paper entitled "Beyond subjective and objective in statistics" to appear in JRSS A
- Boulesteix et al's paper entitled "On fishing for significance and statistician’s degree of freedom in the era of big molecular data" to appear in the philosophy volume "Ott, Max; Pietsch, Wolfgang; Wernecke, Jörg. Berechenbarkeit der Welt? Philosophie und Wissenschaft im Zeitalter von Big Data. Wiesbaden: Springer VS".
- "A critical evaluation of the current p-value controversy" to appear in Biometrical Journal, including discussions by independent authors.

Specific comments:

Abstract:
- "at a realistic statistical power of "40%, given that there is a true effect, only one in six studies will significantly replicate the significant result of another study": I think that this statement should be either explained more or presented without figures.

- "Data dredging, p-hacking and publication bias should be addressed by removing fixed significance thresholds". I would formulate this more cautiously. I essentially agree with the main claim of the authors, but also think that removing fixed significance thresholds will probably not solve *all problems instantaneously*, hence the need for a more cautious formulation.

- The term "p-value hacking" should be defined.

- "But current incentives to hunt for significance lead to publication bias against nonsignificant findings". Publication bias can also be seen as the cause (not (only) as the consequence) of hunting for significance in the sense that "since there is a tendency of journals to publish only/primarily significant results, researchers feel urged to hunt for significance".

Main body:

- p3: "for the following reasons it is actually not" -> I would say "... we claim it is actually not".

- p4: "Second, the claim to remove significance thresholds is widely shared among statisticians". This is obviously an important fact that gives more weight to the authors' considerations, but not really an argument in itself.

- p6: The description of the Cumming study should either include more details or not include any figures at all.

- p7: "Thus, p-values are only as reliable". Why "Thus"?

- p8: "One reason why replication studies are rarely done...". Another one is that there is poor incentive for researchers to perform replication studies. These studies are often not considered as innovative enough by the community. A change of mentalities at all levels (journals, appointment committees, funding agencies, etc.) would be necessary for more replication studies to be performed in practice.

- p8: "There is currently no consensus how replicability or results reproducibility should be measured". An interesting paper on this issue including five case studies is:
https://bmcmedresmethodol.biomedcentral.com/articles/10.1186/1471-2288-8-18

- p9: "However, the above replicabilities of significance are probably overestimated for two reasons." Another major reason is selective reporting (which is discussed only later in the manuscript but could perhaps be mentioned here).

- p10: "In fact, if replication success in terms of statistical significance were larger than just described, this would be highly dubious." What does the word "this" refer to?

- pp10-11: Selective reporting may also occur even if p-value thresholds are not used. While the authors also acknowledge this fact, I think that the manuscript could be a bit misleading in this respect. Naive readers may understand that selective reporting is *mainly/only* due/related to thresholds, which is (at least in my opinion) not the case.

- p18, typo: "Nyman" -> "Neyman"

- p21: Regarding accumulating studies: an important question is whether many independent studies are better or whether a single large study is better. This question is not trivial, not related to statistical testing only and highly context dependent. But it may be worth a remark at this stage (even if it is discussed later).

- p23: "Interpreting larger p-values as graded evidence, but as evidence that *generally speaks against the null hypothesis (perhaps unless p=1)*, would clearly help increasing inferential reproducibility by reducing the choices for qualitative interpretation." I find this statement a bit dangerous, especially for non-statisticians who may not understand it correctly.

- p23: A short introduction to the section "Arguments against removing threshold" may be good.

- For hasty readers, the "arguments against removing thresholds" (which appear as subsection titles) may look like assertions, although they are not intended as such. Please find a way to present these ideas while avoiding confusion.

- The paper needs at least a short conclusion section.

- The paper is quite long and I feel that some redundancies may be shortened (even if it is not strictly necessary from my point of view, considering the atypical character of the paper).

---

## Round 0.2 · accepted · Accept

The authors have addressed the comments sufficiently well. Though some disagreements with Reviewer 2 remain, the paper can still be an important resource for reproducibility research.